# Nitrofurantoin–Aminoglycoside Synergy Against Common Uropathogens Evaluated by Disc Diffusion: A Pilot Study

**DOI:** 10.3390/microorganisms13092117

**Published:** 2025-09-10

**Authors:** Filip Bielec, Monika Łysakowska, Małgorzata Brauncajs, Adrian Bekier, Stanisław Klimaszewski, Dorota Pastuszak-Lewandoska

**Affiliations:** Department of Microbiology and Medical Laboratory Immunology, Medical University of Lodz, 92-213 Łódź, Poland; filip.bielec@umed.lodz.pl (F.B.); malgorzata.brauncajs@umed.lodz.pl (M.B.); adrian.bekier@umed.lodz.pl (A.B.); stanislaw.klimaszewski@student.umed.lodz.pl (S.K.); dorota.pastuszak-lewandoska@umed.lodz.pl (D.P.-L.)

**Keywords:** urinary tract infection, *Escherichia coli*, *Klebsiella pneumoniae*, *Enterococcus faecalis*, amikacin, gentamicin, tobramycin, nitrofurantoin, antimicrobial synergy

## Abstract

The emergence of multidrug-resistant uropathogens requires the development of novel therapeutic strategies. This pilot study assessed the in vitro synergy between nitrofurantoin and aminoglycosides (amikacin, gentamicin, and tobramycin) against three major uropathogens: *Escherichia coli*, *Klebsiella pneumoniae*, and *Enterococcus faecalis*. Ninety clinical isolates were tested using the disk diffusion and double-disk synergy methods. Statistical analysis included Kruskal–Wallis and Mann–Whitney U tests, as well as logistic regression models to assess associations between inhibition zone diameters and synergy occurrence. While synergy was observed in all bacterial species, it was neither universally present nor species-specific. Significant associations were identified between nitrofurantoin inhibition zone size and synergy with amikacin and tobramycin in *E. coli*, and with amikacin in *K. pneumoniae*. In *E. faecalis*, synergy was more likely with larger aminoglycoside inhibition zones, particularly tobramycin. These findings underscore the potential of nitrofurantoin–aminoglycoside combinations in treatment of multidrug-resistant urinary tract infections, while emphasizing the need for further studies incorporating quantitative synergy assays and clinical validation.

## 1. Introduction

Urinary tract infections (UTIs) rank among the most prevalent bacterial infections worldwide and are a major driver of antibiotic prescriptions in both outpatient and inpatient settings [1,2,3]. The increasing antimicrobial resistance (AMR) of uropathogenic bacteria to commonly used antibiotics requires the search for new therapeutic strategies, including those that may enhance treatment efficacy and limit the development of AMR [4,5].

Nitrofurantoin is a broad-spectrum antibiotic active against Gram-negative and certain Gram-positive bacteria and has been used for decades as a first-line agent in the treatment of uncomplicated UTIs [6]. The mechanism of action of nitrofurantoin involves intracellular reduction within bacterial cells to reactive metabolites that damage DNA, RNA, proteins, and cell membranes, leading to a rapid bactericidal effect [7]. However, therapeutic failures due to emerging nitrofurantoin-resistant strains—such as *Escherichia coli, Klebsiella pneumoniae*, and *Enterococcus* spp.—are increasingly reported [8].

Aminoglycosides, e.g., gentamicin, amikacin, are potent bactericidal antibiotics primarily used in the treatment of severe Gram-negative infections [9]. They act by irreversibly binding to the 30S ribosomal subunit of bacteria, resulting in mRNA misreading and inhibition of protein synthesis [10]. Despite their high efficacy, the limitations of aminoglycosides include their toxicity (nephrotoxicity and ototoxicity) and possible risk of AMR development when used as monotherapy [11,12]. Therefore, in clinical practice, aminoglycosides are often administered as part of combination therapy, especially in infections caused by multidrug-resistant (MDR) strains [13].

Antibiotic synergy—where combined agents yield a bactericidal effect surpassing the sum of their individual effects—may stem from complementary mechanisms, increased membrane permeability, or simultaneous targeting of distinct metabolic pathways [14]. Some studies on the synergy of nitrofurantoin with other antibiotics, such as trimethoprim, indicate the possibility of significantly increasing the bactericidal effect against selected strains of Gram-negative species [15]. Beyond direct synergy, repurposing studies have demonstrated that adjunctive agents such as zidovudine boost nitrofurantoin efficacy against MDR *K. pneumoniae*, including in murine UTI and lung infection models, indicating that nitrofurantoin combinations remain a promising avenue in combating MDR pathogens [16]. Although these findings did not include aminoglycosides directly, they reinforce the concept that nitrofurantoin partnerships can yield clinical benefits against resistant Gram-negative organisms [17].

In the case of combination therapy with nitrofurantoin and aminoglycosides, early studies suggest that nitrofurantoin, through destabilization of the bacterial cell membrane and disruption of redox systems [6,7], may enhance the penetration of aminoglycosides into the bacterial cell, potentially leading to a synergistic effect. However, data on the combination of nitrofurantoin with aminoglycosides are scarce and mainly limited to in vitro reports [18]. A study using a *Galleria mellonella* infection model demonstrated synergy between nitrofurantoin and amikacin against 12 MDR uropathogenic *E. coli* (UPEC) strains, with monotherapies ineffective at half clinical doses, while combination therapy significantly improved larval survival [17].

In recent years, the problem of AMR has been increasingly recognized as a critical threat to public health, food security, and sustainable healthcare systems worldwide. The World Health Organization (WHO) ranks AMR among the top 10 global public health threats, while the European Commission has declared it a top-three health priority in the European Union (EU) [19,20]. In response, the EU has developed strategic frameworks to address AMR, including the “EU One Health Action Plan against AMR” and, more recently, the co-funded European Partnership on One Health Antimicrobial Resistance (OHAMR). The OHAMR roadmap underscores the importance of coordinated, cross-sectoral research in human, animal, plant, and environmental health to tackle AMR emergence, improve diagnostics, and provide innovative treatment options [21]. This integrated perspective aligns with the One Health paradigm, promoting research into both social and technical interventions to contain the spread of resistant pathogens [19,21].

In the light of the pressing need for novel therapeutic strategies, particularly in the treatment of UTIs caused by MDR pathogens, combination therapies that exploit antibiotic synergy have gained renewed attention. Combinations such as nitrofurantoin with aminoglycosides represent a potential mean of enhancing bactericidal activity while reducing individual drug dosages and toxicity [22]. Despite promising early findings, such combinations remain underexplored [17].

This study thus complements the OHAMR strategic goals by providing pilot in vitro data on the synergistic activity of nitrofurantoin–aminoglycoside combinations against bacteria isolated from UTIs, supporting the evidence base needed for innovative treatment protocols and contributing to future antimicrobial stewardship. The investigated species—*E. coli*, *K. pneumoniae*, and *E. faecalis*—are among the most frequent etiological agents of UTIs in hospitalized patients [1,2]. The obtained results may form the basis for further preclinical and clinical studies that could lead to the development of new therapeutic regimens based on this combination, potentially improving treatment efficacy and reducing the emergence of AMR among the most common uropathogenic bacteria.

## 2. Materials and Methods

In total, 90 most common uropathogenic strains were tested—30 *E. coli*, 30 *K. pneumoniae*, and 30 *E. faecalis*. All isolates came from our department’s culture collection obtained between July and December 2023 from the microbiology laboratory of Central Teaching Hospital of Medical University of Lodz. All isolates were initially cultured from urine samples of patients with suspected UTI. All bacteria were stored in Viabank storage beads (Medical Wire & Equipment, Great Britain, Corsham, UK) at −80 °C and restored by the 2-times passage on Columbia Agar with 5% sheep blood (Thermo Fisher Scientific, Waltham, MA, USA), 18–20 h at 35 ± 2 °C.

The susceptibility for studied antimicrobials was assessed using disk diffusion method following the European Committee on Antimicrobial Susceptibility Testing (EUCAST) methodology, ver. 13 manual [23]. The antibiotic disks with 100 µg nitrofurantoin, 30 µg amikacin, 10 µg gentamicin, and 10 µg tobramycin were applied to the surface of the Mueller–Hinton II agar plates (Thermo Fisher Scientific, Waltham, MA, USA) inoculated with 0.5 McFarland standard (McF) bacterial 0.85% NaCl suspension, and then incubated for 18–20 h at 35 ± 2 °C. The inhibition zone diameter was measured manually with a caliper according to the EUCAST reading guide for the disk diffusion method [24].

The qualitative occurrence of synergy between nitrofurantoin and aminoglycosides (amikacin, gentamicin, and tobramycin) was assessed using double disk synergy test described earlier by Laishram et al. [25]. The antibiotic disks were applied in pairs 20 mm apart to the surface of the Mueller–Hinton II agar plates (Thermo Fisher Scientific, Waltham, MA, USA) inoculated with 0.5 McF bacterial 0.85% NaCl suspension, and then incubated for 18–20 h at 35 ± 2 °C. Synergy was considered present when distortion or clear enhancement of the inhibition zone toward the adjacent disk was observed. Antagonism was defined as visible reduction or truncation of the inhibition zones between the disks, and no interaction (indifference) was recorded when the zones remained unchanged without distortion or overlap. All assays were performed in duplicate. The inhibition-zone outlines were independently interpreted by four researchers (F.B., M.B., A.B., and S.K.). If discrepancies arose, the double disk synergy test was repeated—moving the antibiotic disks closer together or farther apart—until consensus was achieved [25]. In any remaining disputed cases, the supervisor’s decision (D.P.-L.) was final.

### 2.1. Data Analysis

Statistical analyses were conducted to assess differences in inhibition zone diameters and their association with the occurrence of synergy between nitrofurantoin and aminoglycosides. Given that data distributions were non-normal (as confirmed by Shapiro–Wilk testing), non-parametric tests were applied. The Kruskal–Wallis test was used to compare inhibition zone diameters between bacterial species. The relationships between inhibition zone diameters and the presence of synergy were assessed using the Mann–Whitney U test. To explore factors associated with synergy occurrence, species-specific logistic regression models were developed, incorporating inhibition zone diameters of nitrofurantoin and relevant aminoglycosides as independent predictors. This statistical framework was selected to ensure robust evaluation of both interspecies differences (non-parametric tests) and predictive relationships (logistic regression suitable for binary outcomes). All statistical tests were two-tailed, with a significance threshold set at *p* < 0.05. All analyses were performed using Python (version 3.11), employing SciPy (version 1.11.2) for non-parametric testing and Statsmodels (version 0.14.0) for regression modeling. Visualizations were generated using the matplotlib (version 3.8.0) and seaborn (version 0.13.0) libraries.

### 2.2. Ethical Issues

The study was conducted in accordance with Good Clinical Laboratory Practice Guidelines and the Declaration of Helsinki [26,27]. There was no need to have the consent of the Bioethics Committee to conduct this study, because in the light of the law in force in Poland, the study was not a “medical experiment” (no patient information was included).

## 3. Results

Table 1 presents descriptive statistics for the inhibition zone diameters grouped by bacterial species. *E. faecalis* displayed the largest zones for nitrofurantoin but the smallest for aminoglycosides. Among the Gram-negative rods, *E. coli* generally showed wider aminoglycoside zones than *K. pneumoniae*. Graphical visualizations of these results are shown in Figure 1. The raw data obtained in this experiment are provided in Table A1 included in Appendix A1. The spreads and medians indicated significant inter-species differences in inhibition zones, as confirmed by Kruskal–Wallis tests (Table 2).

Figure 2 illustrates the observed frequency of synergy by bacterial species. Chi-square tests revealed no statistically significant interspecies differences in the occurrence of synergy for any of the drug pairs tested: nitrofurantoin × amikacin (*p* = 0.118), nitrofurantoin × gentamicin (*p* = 0.487), and nitrofurantoin × tobramycin (*p* = 0.295). Thus, no particular type of synergy was significantly more common in any individual species. Importantly, apart from synergy, only “no interaction” was observed across all tested strains and drug pairs; no cases of antagonism were detected in any combination.

### Relationship Between Inhibition Zone Diameter and Synergy

The inhibition zone diameter of nitrofurantoin was strongly correlated with the occurrence of synergy in *E. coli*, particularly in combinations of nitrofurantoin with gentamicin or tobramycin. In the case of *K. pneumoniae*, a significant difference was observed only for the combination of nitrofurantoin with amikacin. For *E. faecalis*, synergy was more frequently observed when the inhibition zones of aminoglycosides were larger. Table 3 presents the dependence between inhibition zone diameter and synergy occurrence. Graphical visualizations of these results are shown in Figure 3.

Logistic regression analysis identified inhibition zone diameter for nitrofurantoin as a significant predictor of synergy occurrence, particularly for *E. coli* when combined with amikacin and tobramycin (*p* = 0.008 and *p* = 0.012, respectively). In *K. pneumoniae*, the only significant predictor was nitrofurantoin inhibition zone diameter in combination with amikacin (*p* = 0.043). No significant predictors were observed for other aminoglycoside combinations in this species. For *E. faecalis*, synergy prediction was significantly associated with tobramycin inhibition zone diameter (*p* = 0.012). Additionally, borderline associations were noted between gentamicin inhibition zone and synergy with nitrofurantoin in *E. coli* (*p* = 0.048) and *E. faecalis* (*p* = 0.076). Figure 4 visualizes the predicted probabilities of synergy across inhibition zone ranges for each bacterial species.

## 4. Discussion

### 4.1. Overview of Study Results

This study comprehensively evaluated in vitro synergy between nitrofurantoin and three aminoglycosides—amikacin, gentamicin, and tobramycin—using disk diffusion-based double-disk synergy tests across three clinically relevant uropathogens: *E. coli*, *K. pneumoniae*, and *E. faecalis*. Synergy occurrence was recorded qualitatively, while inhibition zone diameters served as quantitative susceptibility indicators. To better understand factors influencing synergy, statistical modeling, including Mann–Whitney U tests and species-specific logistic regression models, was performed.

Our findings reveal that the presence of synergy was variably distributed across bacterial species and antibiotic combinations, without a universal pattern favoring any single combination or organism. However, statistically significant associations between inhibition zone diameters and synergy occurrence emerged for selected combinations. Specifically, for *E. coli*, nitrofurantoin inhibition zone diameter was a significant predictor of synergy with both amikacin (*p* = 0.008) and tobramycin (*p* = 0.012), while the association with gentamicin approached significance (*p* = 0.048). In *K. pneumoniae*, only the nitrofurantoin and amikacin combination showed a significant relationship (*p* = 0.043). This suggests that increased susceptibility to nitrofurantoin correlates with a higher likelihood of synergy, at least for Gram-negative rods. For *E. faecalis*, tobramycin inhibition zone diameter emerged as the sole significant predictor (*p* = 0.012), with a borderline result observed for gentamicin (*p* = 0.076).

Despite these associations, logistic regression models predicted relatively low maximum probabilities of synergy, rarely exceeding 60%, even for isolates with the largest inhibition zones. This indicates that while inhibition zone size reflects susceptibility, its predictive value for synergy occurrence remains limited. These results collectively suggest that disk diffusion data alone are insufficient for reliable clinical prediction of synergy, underscoring the complex interplay of bacterial physiology and antibiotic interaction dynamics.

### 4.2. Comparison with Previous Synergy Studies

The study performed by Zhong et al. [17] evaluated synergy between nitrofurantoin and amikacin in MDR uropathogenic *E. coli* using checkerboard and time–kill assays. They found synergistic effects across all 12 strains tested and demonstrated >2 log_10_ CFU/mL reduction in kinetic assays, even at sub-Minimal Inhibitory Concentrations (MIC). This was supported by an in vivo *G. mellonella* model, where combination therapy significantly improved larval survival compared to monotherapy.

Our results visually and statistically align with these findings: nitrofurantoin zone size robustly predicted synergy for nitrofurantoin with amikacin, especially in *E. coli* and *K. pneumoniae*. This supports the concept that increased nitrofurantoin activity enhances membrane permeability or alters transport dynamics, facilitating aminoglycoside entry. Clinical applicability is compelling, given amikacin’s broad Gram-negative coverage and retained activity in ESBL-producing pathogens [13,17].

Although less well-characterized than the nitrofurantoin and amikacin pair, prior studies suggest potential synergy between nitrofurantoin and gentamicin. Abdulkareem et al. [18] reported synergistic effects for nitrofurantoin with gentamicin against 18 MDR *E. coli*, particularly at sub-inhibitory MIC combinations. This resonates with our borderline association in *E. coli* and in *E. faecalis*.

Furthermore, the recent study by Antić et al. [28] reported synergy between nitroxoline and gentamicin in *E. faecalis*, reinforcing a broader theme that nitro-compound-based agents can act synergistically with aminoglycosides. Notably, our findings identify *E. faecalis* as a species in which gentamicin synergy with nitrofurantoin appears plausible.

Synergy between nitrofurantoin and tobramycin is comparatively underreported. Our results provide novel evidence by demonstrating that nitrofurantoin zone size significantly predicted synergy with tobramycin in *E. coli* and in *E. faecalis*. This suggests that akin to amikacin and gentamicin, tobramycin uptake may also be facilitated by nitrofurantoin-induced membrane effects. This has potential clinical relevance in mixed-species or Gram-positive infections where tobramycin is indicated.

Nitrofurantoin’s synergy extends beyond aminoglycosides. Studies with fosfomycin, β-lactams, and non-antibiotic adjuvants show enhanced activity in vitro. These combinations exploit nitrofurantoin’s multiple targets, including ribosomal interference, DNA damage, and metabolic disruptions [29]. Such data support a broader paradigm in which nitrofurantoin acts as an adjunct enhancing membrane permeability or modulating stress responses, thereby potentiating partner antibiotics. While our study focused on nitrofurantoin and aminoglycosides, it aligns with a growing body of literature advocating combination therapy to combat MDR uropathogens [30].

### 4.3. Clinical Implications

The practical relevance of the observed nitrofurantoin–aminoglycoside synergy extends beyond in vitro dynamics. Combination antibiotic therapies are increasingly explored as strategies to overcome MDR, especially in UTI, where both nitrofurantoin and aminoglycosides remain critical treatment options. Our findings contribute to this context by offering insights into how standard susceptibility indicators, like inhibition zone diameters, may help inform decisions regarding such combinations.

Clinicians often rely on aminoglycosides as last-line options in complex urinary infections, where oral therapy is limited. Nitrofurantoin, traditionally reserved for uncomplicated cystitis, shows promise as an adjunct even in resistant infections. Our findings of statistically significant synergy associations in *E. coli* and *K. pneumoniae* further reinforce the rationale for considering nitrofurantoin and amikacin combinations in empirical therapy, particularly when aminoglycoside monotherapy poses toxicity concerns—such synergy could potentially allow for lower amikacin doses while maintaining therapeutic efficacy.

This recommendation may gain additional support from local epidemiological surveillance data. A recent 3-year cumulative antibiogram study conducted at our hospital confirmed persistently high susceptibility levels of *E. coli* and *K. pneumoniae* to amikacin (>90% and >80%, respectively), alongside high susceptibility of *E. coli* and *Enterococcus* spp. to nitrofurantoin (>90%) [31]. These findings validate the empirical efficacy potential of both agents against dominant uropathogens, while being consistent with our observed in vitro synergy. Additionally, due to nitrofurantoin’s established role in treating uncomplicated UTIs and amikacin’s retained efficacy against multidrug-resistant strains—including ESBL-producing Enterobacterales—their combination could bridge therapeutic gaps between oral and parenteral regimens, particularly in complicated infections or when oral therapy alone is insufficient. The local recommendation of nitrofurantoin for uncomplicated UTIs and amikacin for complicated cases reflects this dual applicability.

### 4.4. Strengths and Limitations

A key strength of our study lies in use of standardized disk diffusion methodology in accordance with EUCAST guidelines, allowing reproducible and clinically relevant assessment of antibiotic susceptibility and synergy. Moreover, the evaluation of synergy through double-disk synergy testing, interpreted independently by four experienced researchers, mitigated observer bias and ensured data reliability. The inclusion of three representative uropathogens (*E. coli*, *K. pneumoniae*, *E. faecalis*) broadens the applicability of our findings across both Gram-negative and Gram-positive clinical isolates. A further methodological advantage was the incorporation of species-specific logistic regression models, enabling quantitative analysis of the relationship between inhibition zone diameters and synergy occurrence.

However, several limitations must be acknowledged. First, the sample size (n = 30 per species) restricts statistical power and generalizability, particularly for subgroup analyses. The total number of 90 isolates reflected the full set of consecutive, non-duplicate strains collected during a six-month period at our hospital’s microbiology laboratory. As this project was designed as a pilot study, the sample size was not determined by formal statistical calculation but was based on the practical availability of clinical material. This approach ensured unbiased inclusion of the most common uropathogens encountered in routine diagnostics while allowing generation of exploratory data that may inform the design of larger-scale studies. Second limitation is that reliance on disk diffusion data alone inherently limits the precision of synergy assessment compared to quantitative methods such as time–kill curves or checkerboard MIC assays. Despite these limitations, our study provides important phenotypic evidence supporting the adjunctive potential of nitrofurantoin in combination with aminoglycosides and highlights areas for future investigation, including larger-scale studies, molecular profiling, and in vivo validation.

## 5. Conclusions

This pilot study has demonstrated that nitrofurantoin–aminoglycoside combinations exhibit measurable in vitro synergy against common uropathogens. Inhibition zone diameters, especially for nitrofurantoin, have been found to be significantly associated with synergy occurrence, although with limited predictive value. Importantly, the combination of nitrofurantoin and amikacin appears clinically promising, supported by both our findings and the recent literature, including in vivo synergy models and local antibiogram data showing high susceptibility rates to both agents. Given the rise of multidrug-resistant pathogens and the global challenge of antimicrobial resistance, nitrofurantoin-based combinations may represent a practical, low-cost strategy to enhance aminoglycoside efficacy and optimize UTI management. Further studies, integrating molecular resistance profiling and clinical outcome data, are warranted to validate and refine this therapeutic approach. Larger multicenter studies should combine disk diffusion with complementary synergy assays, such as checkerboard and time–kill methods, while also incorporating genomic analyses to clarify underlying resistance mechanisms. Ultimately, prospective clinical investigations are needed to determine the translational relevance of nitrofurantoin–aminoglycoside combinations in the treatment of MDR UTIs.

## Figures and Tables

**Figure 1 microorganisms-13-02117-f001:**
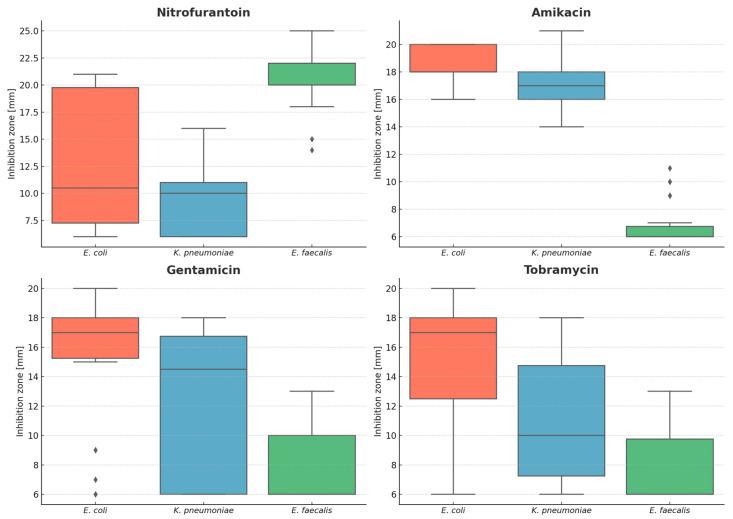
Boxplots for the inhibition zone diameters grouped by studied bacterial species (*Enterococcus faecalis*, n = 30; *Escherichia coli*, n = 30; *Klebsiella pneumoniae*, n = 30). Each boxplot displays the interquartile range (IQR), with the horizontal line representing the median value. Whiskers extend to the most extreme data points within 1.5×IQR from the quartiles, and dots indicate statistical outliers beyond this range.

**Figure 2 microorganisms-13-02117-f002:**
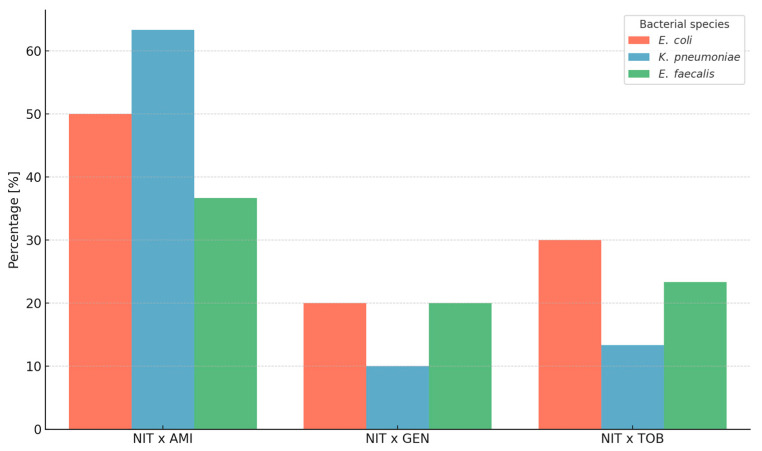
The frequency of observed synergy between nitrofurantoin (NIT) and aminoglycosides (AMI—amikacin, GEN—gentamicin, and TOB—tobramycin) in studied bacterial species (*Enterococcus faecalis*, n = 30; *Escherichia coli*, n = 30; *Klebsiella pneumoniae*, n = 30).

**Figure 3 microorganisms-13-02117-f003:**
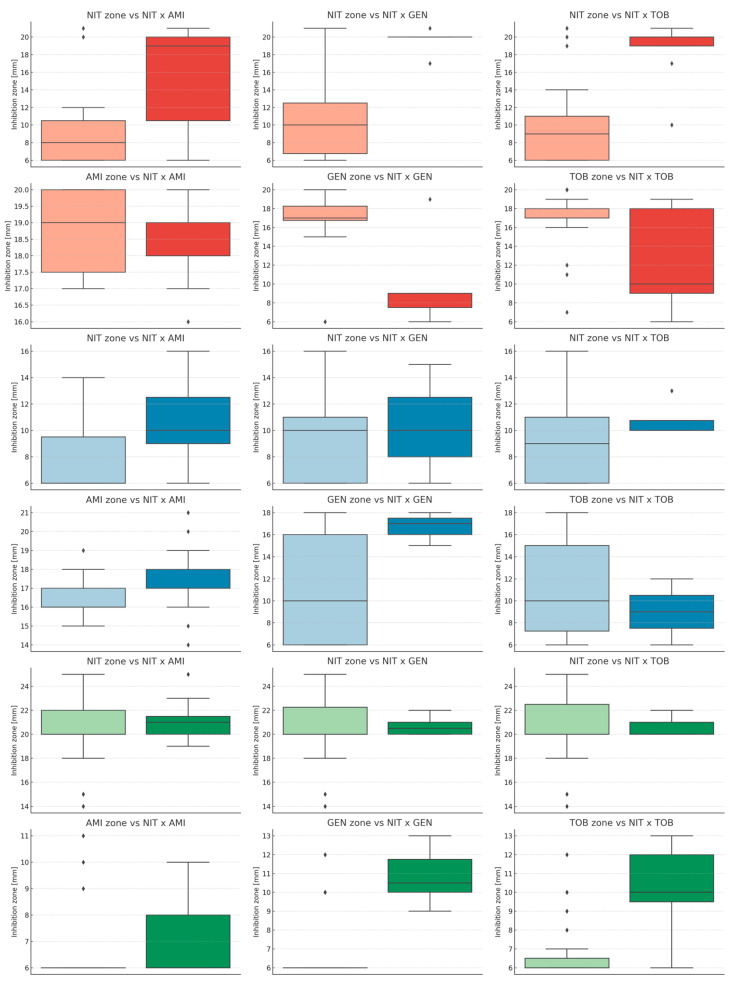
Comparison of inhibition zone diameters in relation to synergy occurrence for each antibiotic (NIT—nitrofurantoin; AMI—amikacin; GEN—gentamicin; TOB—tobramycin) combination grouped by studied bacterial species (red color—*Escherichia coli*, n = 30; blue color—*Klebsiella pneumoniae*, n = 30; green color—*Enterococcus faecalis*, n = 30;). Light-colored boxes represent lack of synergy while dark-colored ones—the occurrence of synergy. Each boxplot displays the interquartile range (IQR), with the horizontal line representing the median value. Whiskers extend to the most extreme data points within 1.5 × IQR from the quartiles, and dots indicate statistical outliers beyond this range. Separate subplots are shown for each species-antibiotic synergy combination.

**Figure 4 microorganisms-13-02117-f004:**
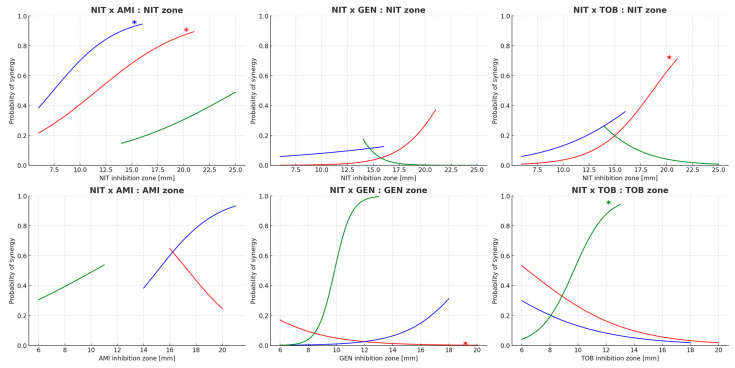
Probability plots from logistic regression models predicting the occurrence of synergy for each antibiotic combination (NIT—nitrofurantoin; AMI—amikacin; GEN—gentamicin; TOB—tobramycin). Top row shows models using NIT inhibition zone as predictor, while the bottom row presents models based on inhibition zones of AMI, GEN, or TOB depending on the synergy combination. Curves are colored by bacterial species (red—*Escherichia coli*, n = 30; blue—*Klebsiella pneumoniae*, n = 30; green—*Enterococcus faecalis*, n = 30). Asterisks indicate statistically significant predictors (*p* < 0.05) for the respective species and predictor variable.

**Table 1 microorganisms-13-02117-t001:** Descriptive statistics for the inhibition zone diameters grouped by studied bacterial species (*Enterococcus faecalis*, n = 30; *Escherichia coli*, n = 30; *Klebsiella pneumoniae*, n = 30).

Antimicrobial	Species	Mean [mm]	Median [mm]	Minimum [mm]	Maximum [mm]	Range [mm]
Nitrofurantoin	*E. faecalis*	20.6	20.0	14	25	11
	*E. coli*	12.7	10.5	6	21	15
	*K. pneumoniae*	9.6	10.0	6	16	10
Amikacin	*E. faecalis*	6.8	6.0	6	11	5
	*E. coli*	18.4	18.0	16	20	4
	*K. pneumoniae*	17.1	17.0	14	21	7
Gentamicin	*E. faecalis*	7.7	6.0	6	13	7
	*E. coli*	15.3	17.0	6	20	14
	*K. pneumoniae*	11.8	14.5	6	18	12
Tobramycin	*E. faecalis*	7.7	6.0	6	13	7
	*E. coli*	15.5	17.0	6	20	14
	*K. pneumoniae*	10.8	10.0	6	18	12

**Table 2 microorganisms-13-02117-t002:** Results of the Kruskal–Wallis tests: interspecies differences in inhibition zone diameters.

Antimicrobial	H Statistics	*p*-Value
Nitrofurantoin	48.47	2.99 × 10^−11^
Amikacin	66.20	4.21 × 10^−15^
Gentamycin	31.61	1.37 × 10^−7^
Tobramycin	40.77	1.40 × 10^−9^

**Table 3 microorganisms-13-02117-t003:** Dependence between inhibition zone diameter and synergy occurrence, grouped by studied bacterial species (*Enterococcus faecalis*, n = 30; *Escherichia coli*, n = 30; *Klebsiella pneumoniae*, n = 30).

Species	Synergy Kind	Inhibition Zone	U Statistics	*p*-Value
*E. coli*	NIT × AMI	NIT	51.5	0.011
		AMI	129.0	0.489
	NIT × GEN	NIT	13.5	0.002
		GEN	116.0	0.022
	NIT × TOB	NIT	22.0	0.001
		TOB	137.5	0.052
*K. pneumoniae*	NIT × AMI	NIT	58.0	0.043
		AMI	66.0	0.089
	NIT × GEN	NIT	37.0	0.832
		GEN	16.0	0.089
	NIT × TOB	NIT	39.5	0.454
		TOB	66.5	0.389
*E. faecalis*	NIT × AMI	NIT	90.5	0.550
		AMI	78.5	0.158
	NIT × GEN	NIT	71.0	0.979
		GEN	12.5	<0.001
	NIT × TOB	NIT	84.0	0.880
		TOB	23.0	0.002

NIT—nitrofurantoin; AMI—amikacin; GEN—gentamicin; TOB—tobramycin.

## Data Availability

The original contributions presented in this study are included in Appendix A. Further inquiries can be directed to the corresponding author.

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
