# Peer review of "Nitrofurantoin–Aminoglycoside Synergy Against Common Uropathogens Evaluated by Disc Diffusion: A Pilot Study"

_microorganisms, 2025, doi:10.3390/microorganisms13092117_

Round 1
Reviewer 1 Report
Comments and Suggestions for Authors
The Manuscript submitted for review raises an interesting topic relevant from an application/medical perspective. In their work, the Authors examined the relationship between nitrofurantoin and aminoglycosides using the disk diffusion method. The studies were performed on the most common strains causing urinary tract infections (Escherichia coli, Klebsiella pneumoniae, and Enterococcus faecalis; a total of 90 clinical isolates).
The Introduction outlined the current research problem and demonstrated the purpose of the Authors’ research. I have no reservations about the methodology, the way the experiments were performed, or the interpretation of the results. However, I have a question regarding the presentation of the results – why did the Authors decide that such a broad statistical approach was necessary in this publication? Are the results of the disk diffusion method alone insufficient to be published as a separate article?
Below are some minor errors:
Line 47: "posible risk" - to correct
Line 106: "urinary tract infection" -
You can use the abbreviation (UTI) because it has already been used before
Line 115: 0.5 McF - provide the full name
Author Response
Comments and Suggestions for Authors
The Introduction outlined the current research problem and demonstrated the purpose of the Authors’ research. I have no reservations about the methodology, the way the experiments were performed, or the interpretation of the results.
1. However, I have a question regarding the presentation of the results – why did the Authors decide that such a broad statistical approach was necessary in this publication? Are the results of the disk diffusion method alone insufficient to be published as a separate article?
Response: We sincerely thank the Reviewer for this thoughtful question. The primary reason for including an extended statistical analysis was to strengthen the interpretability and transparency of our findings. While the disk diffusion and double-disk synergy tests provide valuable qualitative insights, we felt that statistical evaluation would add a quantitative dimension, allowing us to demonstrate possible associations between inhibition zone diameters and the occurrence of synergy. This approach, in our view, increases the robustness of the study and facilitates comparisons with other reports in the literature, where statistical modeling is often employed to evaluate antimicrobial interactions. We fully agree that the raw results of the disk diffusion method are already meaningful, yet we believe that complementing them with additional statistical tools provides readers with a deeper understanding of the data and may serve as a useful reference for future studies.
2. Reviewer comment: Below are some minor errors:
Line 47: "posible risk" - to correct
Line 106: "urinary tract infection" -
You can use the abbreviation (UTI) because it has already been used before
Line 115: 0.5 McF - provide the full name
Response: All minor errors were corrected.
Reviewer 2 Report
Comments and Suggestions for Authors •The manuscript does not indicate how the sample size was determined. •The justification for the choice of statistical analysis methods is not provided. •A demographic table describing the study population is missing, and there is no analysis of the correlation between patients’ vital signs and their investigative findings in relation to bacterial count. •The authors should clarify the legal framework in Poland that allows exemption from bioethics committee approval despite the use of biological material. •The conclusion section is weak and does not include recommendations for future research or directions for further studies.Author Response
We thank the Reviewer for pointing out all issues below.
Reviewer comment no. 1.
The manuscript does not indicate how the sample size was determined.
Response: We thank the Reviewer for pointing out this issue. The sample size of 90 isolates (30 E. coli, 30 K. pneumoniae, and 30 E. faecalis) was determined by the availability of clinical strains collected during six-months at our hospital’s microbiology laboratory. Since this study was designed as a pilot investigation, our primary aim was not to achieve statistical representativeness but to provide exploratory in vitro data that may guide future, larger-scale studies. We fully acknowledge this limitation and have now clarified the rationale for our sample size in the revised version of the manuscript (Strengths and Limitations section).
Reviewer comment no. 2. The justification for the choice of statistical analysis methods is not provided.
Response: In the revised version of the manuscript, we have clarified the rationale for our statistical approach within the Data Analysis section. Specifically, we now state that non-parametric methods (Kruskal–Wallis and Mann–Whitney U tests) were applied because the data did not follow a normal distribution (as confirmed by Shapiro–Wilk testing). In addition, logistic regression was employed as it is suited well for binary outcomes such as the presence or absence of synergy. This explicit explanation has been incorporated into the text to ensure that the choice of statistical methods is fully transparent and appropriate to the dataset.
Reviewer comment no. 3. A demographic table describing the study population is missing, and there is no analysis of the correlation between patients’ vital signs and their investigative findings in relation to bacterial count.
Response: We would like to emphasize that our study was designed as a laboratory-based, in vitro investigation. The analysis was performed exclusively on bacterial isolates obtained from routine urine cultures, without access to patient-identifiable data, vital signs, or clinical records. Consequently, no demographic or clinical variables were collected, and the study population consisted of microbial strains rather than patients. For this reason, the inclusion of a demographic table or clinical correlation analysis was not applicable.
Reviewer comment no. 4. The authors should clarify the legal framework in Poland that allows exemption from bioethics committee approval despite the use of biological material.
Response: We thank the Reviewer for raising this important issue. Our study was conducted exclusively on anonymized bacterial isolates obtained from routine urine cultures. According to Polish law [Act of 5 December 1996 on the Professions of Physician and Dentist - Journal of Laws 2024, item 1287, as amended], information or material derived from medical procedures may be used for scientific purposes without the consent of participants provided that individual identification is impossible, and the collection or use of biological material for scientific purposes does not constitute a medical experiment. Consequently, this type of laboratory-based in vitro research does not require the approval of a bioethics committee.
Reviewer comment no. 5. The conclusion section is weak and does not include recommendations for future research or directions for further studies.
Response: In the revised version of the manuscript, we have expanded the Conclusion section to provide clearer recommendations for future research.
Round 2
Reviewer 2 Report
Comments and Suggestions for Authors
Thanks for doing modification